# Benefit-cost analysis of coordinated strategies for control of rabies in Africa

A. Bucher[1,2,5], A. Dimov [1,2,3,5], G. Fink[1,2], N. Chitnis[1,2], B. Bonfoh[4] & J. Zinsstag [1,2] ✉

Previous research suggests that dog mass vaccination campaigns can eliminate rabies locally, resulting in large human and animal life gains. Despite these demonstrated benefits, dog vaccination programs remain scarce on the African continent. We conducted a benefit-cost analysis to demonstrate that engaging into vaccination campaigns is the dominant strategy for most countries even in the absence of coordinated action between them. And quantify how coordinated policy measures across countries in Africa could impact rabies incidence and associated costs. We show that coordinated dog mass vaccination between countries and PEP would lead to the elimination of dog rabies in Africa with total welfare gains of USD 9.5 billion (95% CI: 8.1 – 11.4 billion) between 2024 and 2054 (30 years). Coordinated disease control between African countries can lead to more socially and ecologically equitable outcomes by reducing the number of lost human lives to almost zero and possibly eliminating rabies.

Rabies remains a neglected disease and persistent human and animal health problem throughout the low and middle income countries[1]. Rabies is a fatal neurological pathogen primarily spread by dogs, and is estimated to cause ~59,000 human deaths, 3.7 million disability-adjusted life years lost and USD 8.6 billion economic losses annually[2] and at least 250,000–500,000 dog deaths[1,3,4]. Due to successful anti-rabies campaigns in the past decades, Dog-transmitted rabies is unequally distributed, with the low and middle income countries carrying 99% of the burden[5]. The World Health Organization (WHO) aims at zero human deaths from dog-mediated rabies by 2030[6]. Human deaths from rabies can be prevented through timely administration of post-exposure prophylaxis (PEP) for people who have been bitten. However, PEP availability remains scarce in most local healthcare systems and PEP compliance is poor and dog vaccination is limited.

While the use of PEP alone cannot address dog-mediated rabies[7], in principle dog-mediated rabies can be controlled and eliminated by the mass vaccination of dogs if a sufficiently high coverage can be achieved[8]. A One Health (OH) approach aims to demonstrate an incremental benefit from a closer cooperation between human and animal health that cannot be demonstrated if the sectors work in isolation[9,10]. Compared to PEP, dog mass vaccination has the main advantage that it has the potential to permanently interrupt transmission of rabies and to also prevent millions of unnecessary human and animal deaths[8,11]. The main challenges with dog rabies vaccination campaigns are access to vaccines, the high mobility among dogs, mostly through human-mediated dog transport[12,13], resulting in an almost certain pathogen reintroduction of rabies at the local and national levels[14,15]. Large-scale synchrony of canine rabies incidence in Africa strongly indicates cross-border transmission of dog rabies[1]. Active protection of the country's borders could avoid the reintroduction from the outside through direct dog-to-dog contact and human-mediated dog transport in principle[16], but is likely not feasible across countries.

The constant threat of pathogen importation means that the impact of policy actions taken by each country becomes dependent on the efforts made by others. Such strategic policy choices can be analyzed through a mathematical framework (game theory). In such a framework, we monetize the outcomes of different policies for all interacting actors; that way, we can compare them to find the most profitable self-interested or cooperative choices. In this paper, we

[1]Swiss Tropical and Public Health Institute, Kreuzstr. 2, 4123 Allschwil, Switzerland. [2]University of Basel, Petersplatz 1, 4003 Basel, Switzerland. [3]University of Zürich, Rämistrasse 71, 8006 Zürich, Switzerland. [4]Centre Suisse de Recherches Scientifiques en Côte d'Ivoire, 01 BP 1303 Abidjan 01, Côte d'Ivoire. [5]These authors contributed equally: A. Bucher, A. Dimov. ✉e-mail: jakob.zinsstag@swisstph.ch

present the economic and health impacts at the country level of various combinations of human post-exposure prophylaxis and dog rabies vaccination programs throughout the African continent. The goal is to compare how and how much inter-countries cooperation and intersectoral cooperation affect subsequent health, financial and environmental outcomes[17]. We use this approach to investigate and illustrate the strategic dilemma underlying current policies and the potential benefits of reaching a socially optimal policy equilibrium through cooperation[18].

## Results

### Extrapolation of the transmission dynamics model

Our model suggests a total canine population of 141,398,505 (95% CI: 102,147,668–195,460,271) across the 48 countries included in our study for the base year 2024. For different strategy profiles, country-level payoffs are presented in the Supplementary Data 1.

Our Susceptible-Exposed-Infectious-Removed (SEIR) model suggests a total of 15,275,368 (95% CI: 9,606,726–23,718,481) humans exposed to rabid dogs for the baseline between 2024 and 2054. Thus leading to an estimate of 2,902,293 clinical cases without PEP (95% CI: 1,796,748–4,555,867) and a number of human lives lost around 459,068 (95% CI: 277,794–732,116) assuming estimated incomplete PEP reach numbers as previously published by Hampson[2].

Figure 1 illustrates the total number of Humans Exposed to Rabid Dogs for all 48 countries in scope, for some strategy profiles. (1) Strategy profile "Baseline" corresponds to the status quo where only incomplete PEP administration is used. (2) Strategy "Vaccination campaign in all countries" assumes PEP administration as well as two consecutive vaccination campaigns in all countries in the years 2024 and 2025, thus interrupting rabies transmission as rabies is eliminated within the animal reservoir. The other strategy profiles are mixed profiles, where some countries stay with PEP administration, while others mass vaccinate. In those cases, rabies is not eliminated and is reintroduced continuously from outside the respective countries. As a result, the yearly incidence of humans exposed to rabid dogs increases

again until eventually reaching the endemic equilibrium. We illustrated this trend for the following strategy profiles: (3) one defecting country, (4) with 50-50 distribution of vaccinating and (6) Nash equilibrium, countries with vaccination as dominant strategy vaccinate, whereas the others don't.

### Strategy economic valuation

The average estimated cost per vaccinated dog was 4.47 USD, ranging from 1.8 USD (95% CI: 1.46 USD–2.11 USD) for Burundi and 13.1 USD (95% CI: 10.12 USD–16.54 USD) for Equatorial Guinea per canine. The country-specific average number of rabies-induced years of life lost (YLL) in the working age (16–64 years of age) ranges between 23.5 years (Tunisia) and 26.4 years (Guinea and Uganda) with a median value of 25.8 years (mean = 25.6 Years). For the base year 2024, the country-specific capital loss of one life lost due to rabies is valued between USD 6764.04 (Burundi) and USD 238,904.47 (Equatorial Guinea), with a median value of USD 32,993.13 (mean = USD 60,272.76).

Excluding the human capital effect (HCE) cost from the payoff is equivalent to comparing the costs of vaccination and PEP use. Considering the same strategy profiles as before and the aggregated payoff over all 48 countries, the breakeven point between the baseline strategy (1) and the coordinated mass dog vaccination (2) is in 2033. In comparison, the other strategies are less profitable than the baseline strategy (cf. Supplementary Fig. 5 in Supplementary Information 2). If we take into account the human capital effect, the breakeven point shifts to 2025. For some countries, the HCE gains are so important that it is profitable since the beginning of the vaccination campaign. We can also observe that overall all strategies become more profitable than the baseline (cf. Supplementary Fig. 6 in Supplementary Information 2). In a more detailed view of the difference between the payoffs of the baseline strategy and coordinated mass dog vaccination strategy per country and year (cf. Supplementary Fig. 7 in Supplementary Information 2). If we consider the payoff difference without HCE, and we observe that some countries don't have any breakeven point during

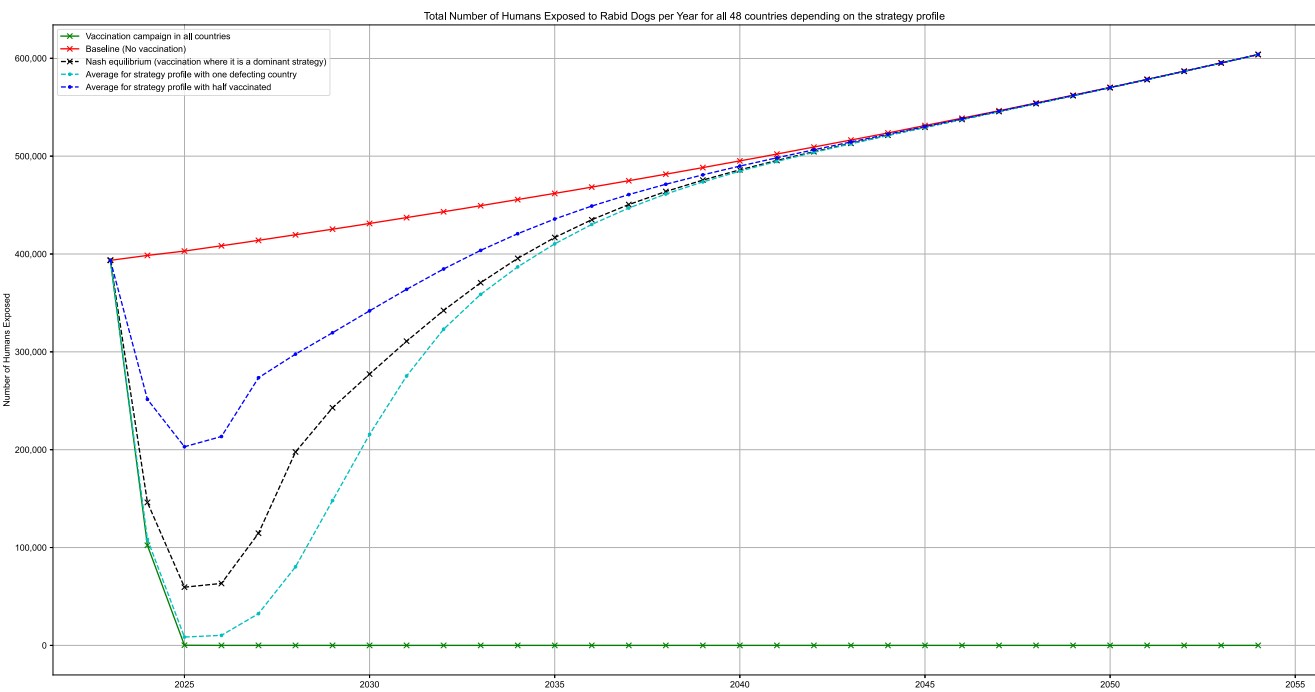

**Fig. 1 | Yearly human exposure to rabid dogs all 48 countries for different strategy profiles at the end of each year between 2024–2054 (30 years).** Baseline strategy (i) in red, full cooperation (ii) in green, one defecting (iii) in cyan, half mass vaccinate (iv) in blue, Nash equilibrium (v) in black.

**Table 1 | Average payoff difference (compared to the baseline) by strategy profile for each country in USD between 2024–2054 (30 years)**

| Group | Country code | Payoff by strategy profile (in USD) | | |
|---|---|---|---|---|
| | | All PEP | All VAC | One VAC[a] |
| Vaccination dominant (Group 1) | AGO | – | 166,398,474 | 24,627,784 |
| | BDI | – | 77,279,689 | 14,398,174 |
| | BEN | – | 97,435,632 | 17,419,729 |
| | BFA | – | 328,590,597 | 66,338,594 |
| | CAF | – | 41,146,977 | 7,635,309 |
| | CIV | – | 448,942,753 | 85,478,722 |
| | CMR | – | 113,500,806 | 17,735,038 |
| | COD | – | 1,315,179,719 | 246,432,786 |
| | COG | – | 14,952,471 | 1,697,740 |
| | DJI | – | 10,238,660 | 2,059,611 |
| | ERI | – | 27,292,770 | 5,622,705 |
| | ETH | – | 1,467,340,154 | 289,387,321 |
| | GHA | – | 85,174,192 | 9,235,401 |
| | GIN | – | 211,824,911 | 43,109,473 |
| | GMB | – | 11,046,401 | 2,058,703 |
| | GNB | – | 18,954,824 | 3,825,930 |
| | GNQ | – | 11,500,025 | 1,178,091 |
| | KEN | – | 227,908,213 | 33,038,023 |
| | LBR | – | 45,812,600 | 9,425,550 |
| | LSO | – | 9,742,964 | 1,900,394 |
| | MAR | – | 57,608,893 | 694,421 |
| | MLI | – | 246,230,011 | 45,851,678 |
| | MOZ | – | 292,320,445 | 56,778,172 |
| | MRT | – | 27,913,591 | 4,777,771 |
| | MWI | – | 129,963,798 | 23,586,652 |
| | NER | – | 491,504,203 | 82,059,249 |
| | NGA | – | 1,369,361,279 | 238,799,543 |
| | RWA | – | 77,235,042 | 13,820,535 |
| | SDN | – | 265,782,099 | 48,869,738 |
| | SEN | – | 98,881,606 | 16,429,966 |
| | SLE | – | 57,586,442 | 11,976,788 |
| | SOM | – | 180,310,968 | 34,466,355 |
| | SSD | – | 72,305,189 | 14,596,629 |
| | SWZ | – | 7,944,025 | 1,050,506 |
| | TCD | – | 237,459,609 | 44,097,694 |
| | TGO | – | 43,257,880 | 7,767,850 |
| | TZA | – | 299,784,026 | 47,831,155 |
| | UGA | – | 227,375,524 | 38,918,723 |
| | ZMB | – | 118,634,683 | 21,167,093 |
| | ZWE | – | 264,741,978 | 53,277,009 |
| No dominant strategy (Group 2) | BWA | – | 3,398,395 | **−992,567** |
| | DZA | – | 19,077,016 | **−11,494,249** |
| | EGY | – | 130,579,996 | **−27,208,448** |
| | GAB | – | 1,517,578 | **−1,077,479** |
| | LBY | – | 468,798 | **−3,247,023** |
| | NAM | – | 3,723,398 | **−559,172** |
| | TUN | – | 4,656,668 | **−3,514,844** |
| | ZAF | – | 87,392,574 | **−18,025,497** |

Negative payoff values are highlighted in bold and justify that these countries are part of Group 2 (without a dominant strategy). "All PEP" all countries stay at baseline to continue to react to human cases using PEP; "all VAC" all countries collaborate to realize a mass dog vaccination program; "one VAC" only the studied country mass vaccinates dogs meaning that we have a pathogen reintroduction in the midterm.

[a]Only the value of the studied country from the profile where it mass vaccinates is reported.

the simulated period. Meaning that even if everybody cooperates, it will not be profitable for them to do it. Nevertheless, by including the human capital effect, it becomes beneficial for everyone in the simulated period. Moreover, for 18 countries (38%), it is profitable since the beginning of the dog rabies vaccination program, while after 4 years, 39 countries (81%) benefit from the program.

## Strategy analysis

By comparing the selected strategy profiles, we found two groups of countries (cf. Table 1). The first group with countries for whom it is always beneficial to mass vaccinate dogs independently of the choices of other countries. The second group comprises countries for whom it is beneficial to mass-vaccinate dogs only if everyone realizes mass-dog vaccination campaigns. We consider only the average values, as it is equal to the expected utility of the strategy.

The strategy profile with countries of group 1 vaccinating dogs and countries of group 2 continuing using only PEP corresponds to the profile with the best expected payoffs for all countries regardless others' strategies (Nash equilibrium in a non-cooperative setting). This means that without cross-country collaboration, the countries of Group 1 will reap the supplementary gains of the dog vaccination campaign reported in the column "One VAC" of Table 1; meanwhile, the countries of Group 2 will stay at the baseline without complementary gains nor losses. With the uncertainty on whether everybody will comply with the dog vaccination campaign or without coordination, this decision profile represents the best choices for all countries. The cooperation profile ("all VAC") holds the greatest average payoff for all countries (Pareto-optimal solution). It also means that if a country changes its choice, all other countries will see their payoff decrease. If we consider a cooperative game with group 1 and group 2 as coalitions, or if we consider this as a repeated game, the highest possible gains for all countries become also the best strategy for all of them (The Pareto-optimal solution becomes a Nash equilibrium). Thus, cooperation is the most rational choice. In addition, a long-term rabies-free state for the African mainland as a common good can only be attained if all countries cooperate (see Supplementary Information 2 for proofs).

By the year 2054 (before the African Union Agenda 2063), the gain from a nationwide coordinated vaccination campaign resulting in an elimination of rabies within the animal reservoir will lead to a total gain of 9547 m USD (95% CI: 7990–11,432 m USD) when compared to the baseline.

In our model, the variable having the most significant impact on the payoff variance is the uncertainty on the exposure with total-order Sobol indices around 0.65 [Q25–Q75: 0.58–0.76]. After that, the probability of receiving PEP with a mean value of total-order Sobol indices about 0.16 [Q25–Q75: 0.01–0.23], and for the initial dog population, the indices are around 0.14 [Q25–Q75: 0.10–0.17]. We have almost the same values for the first-order and total-order Sobol indices, meaning that almost no variance comes from the interactions of the studied variables. We mention that for countries with a high probability of receiving PEP (>99%), the variance of the distribution was reduced. For those countries, we can see that the human capital effect is reduced, so the impact of the vaccination price and the initial dog population are on par with the exposure uncertainty in the case of the pathogen reintroduction (see Supplementary Information 1 and Supplementary Data 3 for details). This analysis also shows that because of the uncertainty on the exposure, the vaccination campaign holds substantially more benefits if the real exposure is underestimated, i.e., dogs bite more than 2.3 times a year. On the other side, the closer PEP availability is to 100%, the more the situation becomes dependent on PEP prices and vaccination prices.

To provide a more granular overview of the results, we hereafter show the incremental gains between the mass dog vaccination strategy

Gains in USD from the cooperation for mass dog vaccination

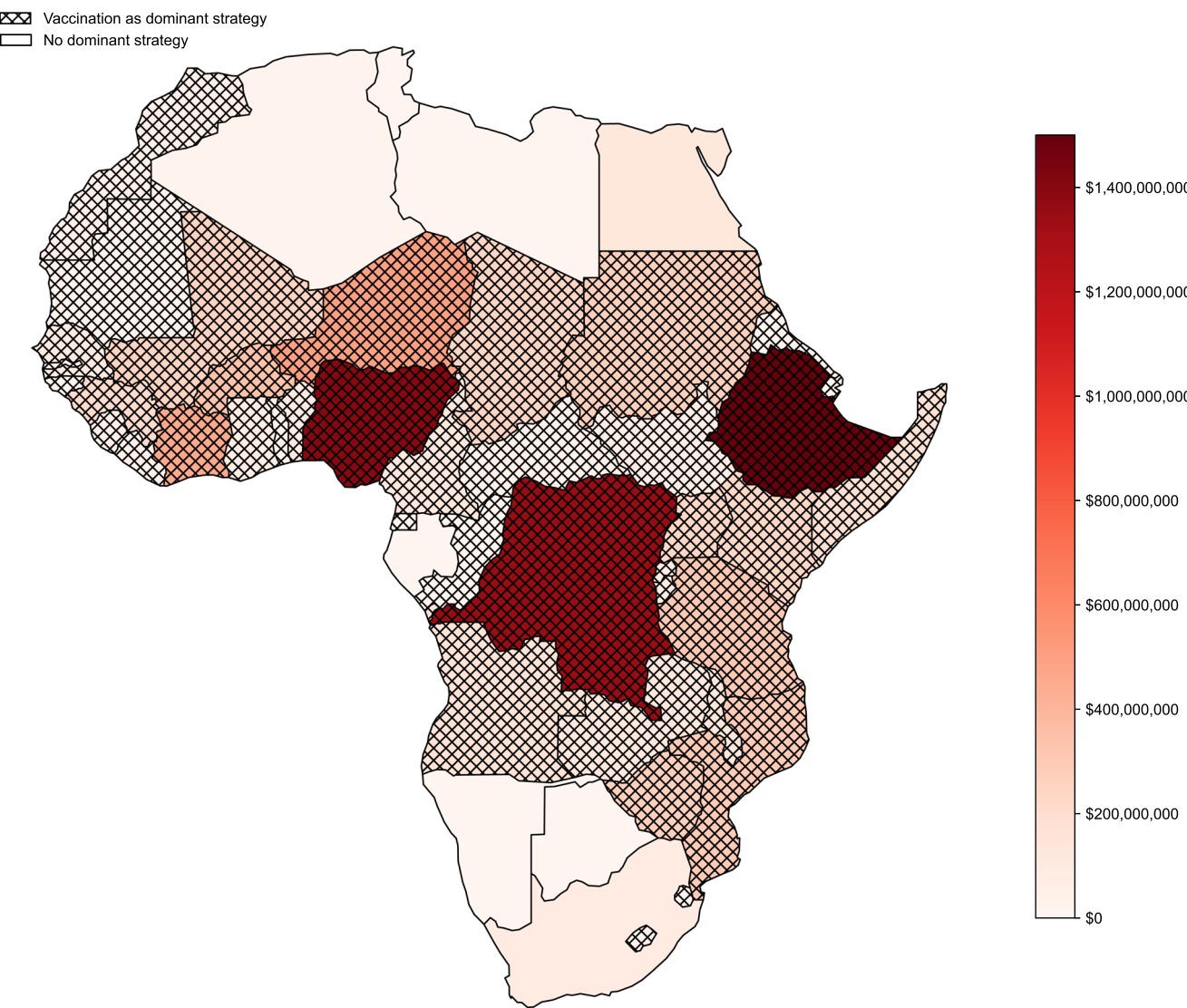

**Fig. 2 | Absolute gain in USD per country comparing the cooperative profile to the baseline.** As well as, the division between the countries with vaccination as dominant strategy and the others between 2024–2054 (30 years).

and the baseline, including the HCE, both as total values over the total 30 years (Fig. 2) as well as relative to the current gross domestic product (GDP) (Fig. 3).

## Discussion

In this article, we have demonstrated that lacking cooperation across countries results in systematic failure to engage in disease control similar to the behaviors in Hardin's seminal work on the "the tragedy of the commons"[19]. In the specific case of rabies, an environment where no human lives are lost due to canine-transmitted rabies and no animal is affected by this disease is clearly feasible[20] using a OH approach[17]. The problem is that the situation currently faced by countries resembles a classic prisoner's dilemma. If every country would act in the interest of the global objective of eliminating rabies in Africa, everybody would reap the health and financial benefits of joint interventions. As previous research has focused on a cost comparison of the interventions of a continuous supply of PEP or the execution of a nationwide canine vaccination campaign, only by taking the monetization of YLLs into account make the execution of a coordinated national dog mass vaccination the superior choice for most of the countries and make the benefits timely. We intentionally simplify the

underlying complex problems of rabies elimination in Africa to identify how the consideration of external effects in the form of monetized benefits can influence the strategic decisions by interdependent actors, to reach the societal optimum. As in most African countries, the public and animal health sectors don't communicate, human capital benefits are not considered related to dog mass vaccination campaigns. Nevertheless, mass dog vaccination is the dominant strategy for most countries and is the strategy holding the greatest benefits irrelevant to the strategies of other countries.

The required regional collaboration to achieve sustained rabies elimination can inspire from similar experiences in the elimination of fox rabies in Europe through centralized financial incentives by the European Union[14], or the coordinating role of the Pan American Health Organization (PAHO) in the elimination of dog rabies in Latin America[21]. The successful Rinderpest elimination in Africa by 2011 considered "the legitimate need of each stakeholder group and the power relationships"[22] to translate the willingness to cooperate at the national, district/province, and the household level through public-private-community partnership. Such social innovations are part of an transdisciplinary OH approach[23] and can enable the realization of the benefits through cooperation at the human-animal interface on a

Gains in % of the GDP from the cooperation for mass dog vaccination

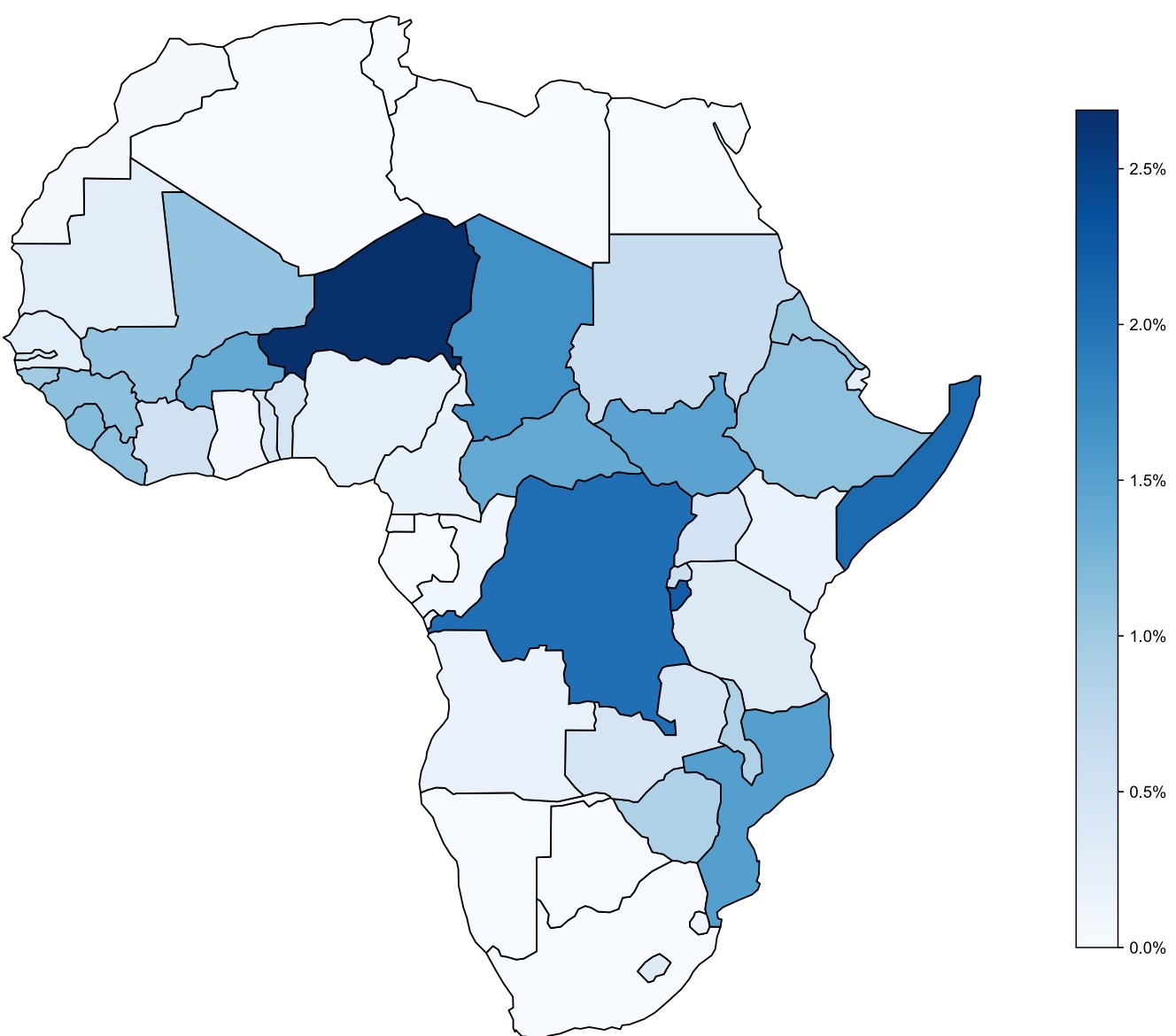

**Fig. 3 | Relative gain in billion USD per country as a percentage of the corresponding GDP.** This figure represents the difference between the cooperative profile's and the baseline's payoffs between 2024–2054 (30 years) in % of the expected 2024 GDP.

country level that otherwise remains unrealized. Further research should focus on this ecology of games[24]. We provide a brief discussion of the different layers of the underlying games in Supplemental Information 3. The ongoing establishment of national One Health platforms in Africa and the recent ECOWAS initiative on the Regional One Health Coordination mechanism are good avenues of applying coordination models. Rabies has been identified by most African countries as a priority disease to be controlled through One Health platforms. The game theory comes timely with these emerging countries' One Health platforms[25,26]. The situational analysis of the collaboration and coordination between sectors and countries revealed the need for multiscale political commitment based on robust economic incentives evidence of One Health. The effective collaboration relies on the institutions and countries' governance, the capacity and knowledge in coordination, the transformed institutional and legal frameworks for complex health problem such as Rabies which is embedded in different ministries.

Acknowledging that the scaling up the initial results of the SEIR model parameterized to the N'Djamena setting has its limitations, we have chosen to follow similar research that has built their continental models upon limited available data[2,27]. Following the suggestion from Anyiam et al., we assume that a country's whole dog population can be vaccinated within 1 year, which is necessary for rabies elimination due to the short average canine live-span[28]. As suggested by Mindekem et al., we emphasize the need for two consecutive vaccination campaigns, each with a reach of 70%, to eliminate rabies[11]. We based our model on the currently available data and acknowledge that rabies elimination might well take up to 10 years within a country. For our analysis, this would mean that human exposure would increase, thus reducing the overall benefit but not the utility ranking. A careful planning of a coordinated Africa-wide campaigns could be achieved through the well-established rabies vaccine bank at the World Organization of Animal Health (WOAH) and a good accessibility framework by countries as implemented by the COVAX initiative. The coordination mechanism in

the strategic option implies a strong link between the PVS and the IHR, the legitimacy of the regional economics regions, mutual learning in the health systems[10].

PEP costs were assumed to follow the same distribution across all countries in scope. As this research is not targeted at deriving exact costs for intervention but on estimating utilities for a game-theoretical discussion, we were willing to take this approach as a starting point and expect heterogeneity of the underlying input parameter to be included in further research. Nevertheless, from our sensitivity analysis, we can argue that if the actual distributions and the ones used are not significantly different (i.e., means outside of the considered price interval), the effect on the model would be marginal. Not all potential associated costs were included regarding monetization, but neither were all benefits. On the cost side, we have solely included the public costs for the PEP, assuming free public availability of PEP. This is the case for some countries like Mali but not for all. The costs for PEP administration would be higher if the opportunity costs of the personnel costs and storage would have been included. The benefit side captures only the direct effects of avoided YLL under GDP contribution and does not consider any additional effects like avoided preventive vaccination. Not including the full social and economic benefits results in underestimating the benefits. We are confident that our estimates are conservative, and the total benefit will likely be higher.

Canine background vaccination at the household level does not reach sufficiently high levels to interrupt transmission due to financial constraints or lack of knowledge. Additionally, canine vaccination is not endorsed by the public health authorities as they see their responsibility in administering PEP. As the veterinary authorities do not see significant gains from vaccinating the canine population, an intervention from that side is not implemented either. The costs of rabies prevention through canine vaccination seem to exceed the status quo costs for any involved party. Through our work we add an additional overarching dimension to the integrated OH approach as suggested by Scoones et al. combining process, pattern and participation[29].

Large-scale synchrony of dog rabies incidence in Africa and the importance of intervention responses suggest that control of canine rabies in Africa will require sustained efforts coordinated across political boundaries[1]. If countries take the human capital loss into consideration African inter-governmental platforms like the African Union Inter-African Bureau for Animal Resources (AU-IBAR) together with the African centre of Disease Control (African CDC) and the Economic Community of West African States (ECOWAS) One Health Coordination mechanisms composed of both West African Health organization (WAHO), the Regional Animal Health Centre (RAHC) and the Department of Environment (DoE) are best placed to achieve such coordination[30]. In this way, responding to the Lancet's call for more ecological equity through a One Health approach[31], the African continent could materialize these incredible financial gains and avoid the continuous loss of numerous human and animal lives, and eventually eliminate rabies.

## Methods

We adapted and extrapolated an mathematical model (deterministic meta-population model) from Laager et al. for N'Djamena, Chad[15]. This model allows for pathogen reintroduction and estimated country-specific dog-rabies cases incidence. The parameters of the Susceptible-Exposed-Infectious-Recovered (SEIR) model are derived from recent publications from the same context[11,15,32] and can be found in Supplementary Table 3 of the Supplementary Information 0. The underlying assumption for the parameters of the model is the homogeneous endemic dog-rabies cases incidence of 1.9 per 100,000 dogs per week over all 48 countries. The number of dogs per country has been derived from utilizing the dog: human ratio for Africa as published by Knobel et al. (cf. Supplementary Data 7 for the calculations) with a distinction between urban and rural populations[27]. The human

population distinguished by rural and urban was derived from the World Bank and the total population as well as the population growth rate by the United Nations World Population Prospects 2019[33].

Using all currently available data (cf. Table 2 for a quick view of different country-specific values and their sources), we modeled and monetized the benefits and costs of various rabies strategies for all countries on mainland Africa. Specifically, we estimated the benefits and cost ("payoffs") for the following three scenarios. Firstly, a baseline strategy under the assumption of no collaboration between the public health and veterinary authorities, focusing solely on incomplete PEP administration (status quo). PEP administration is incomplete because of inadequate supplies and low compliance of patients[34]. Secondly, an uncoordinated national level dog vaccination campaign, with medium term pathogen reintroduction from outside the country border. Thirdly, a coordinated regional vaccination campaign, with coordinated efforts both within countries (public health and veterinary offices collaborating in a OH approach), and cooperation between countries.

To compute the financial implications of various strategies, we compute total life time income losses for each person dying of rabies. Specifically, we assume that each person works ages 15–64, and earns average country-specific wages during this period in terms of gross domestic product (GDP) per capita. The combination of the years of life lost with GDP contribution derived from the age distribution with the country-specific GDP contribution results in the weighted average years of live lost (YLL) with GDP contribution for one average bite victim in a specific country. The details of this methodology are presented in Supplementary Information 1 in the "Economic valuation" section. The countries' GDP per capita was then utilized as a proxy for the economic contribution of an individual in a given year[35]; these values were obtained from the World Bank database on GDP per capita in current US$[36] except for Eritrea and South Sudan, where the values were unavailable and attained from the International Monetary Fund[37].

As an extension of the previous work by Anyiam et al.[28] we calculate the intervention costs of each strategy covering the PEP costs for treating bite victims of suspected rabid dogs according to the estimated PEP reach numbers as previously published by Hampson[2] (cf. Supplementary Data 8 for the original numbers). The original data from Hampson covered Sudan without distinction between North and South. To make this distinction, the bite incidence and death rate were considered the same, and the populations were devised proportionally to 2024 population data.

Each country has a choice between two strategies: only reacting to human cases using PEP administration (baseline) or combining it with a mass dog vaccination campaign. The total cost of the strategy depends on (1) the vaccination cost, (2) the post-exposure prophylaxis cost and (3) the human capital effect cost. Pairing the resulting number of rabid dogs from the SEIR model with a multiplication factor of 2.3 (the exposure factor in our model) in the same geographical location from the 2013 animal bite survey on animal bite injuries[38] estimates the number of exposed humans. For the post-exposure prophylaxis cost, the number of exposed humans must be multiplied by a factor of 1.5 to prevent all human lives from being lost, as ~2/3 of human suspected rabies exposure cases are actually at risk of developing clinical rabies[39,40]. This follows from the logic that our SEIR model calculates the number of humans a rabid dog has truly bitten. As for the human capital effect cost calculation, we consider only 19% of the exposed humans to develop clinical rabies[41], which we combine with incomplete PEP reach and estimated GDP contribution. The details of the methodology and exact formulas can be found in Supplementary Information 1.

As each country can make a different choice, we have a large number of possible combinations. Although, for our analysis, only some are sufficient. Therefore, we will consider the following strategy profiles (i.e., combinations of choices made by all countries): all countries stay at baseline to continue to react to human cases using

**Table 2 | Summarized country-specific input data for the SEIR model, the payoffs calculation in the average case, and some complementary data, such as GDP per capita[36]**

| Country name | ISO3 country code | Dog population | Dog population increase | Probability of receiving PEP | Vaccination price in 2024 | PEP price in 2024 | GDP per capita in 2024 | Human population in 2024 (in $M) | GDP in 2024 (in $B) | GDP contribution in 2024 (in $k) | Deaths (Hampson et al.[2]) | Exposures (Hampson et al.[2]) |
|---|---|---|---|---|---|---|---|---|---|---|---|---|
| Algeria | DZA | 3,252,270 | 0.48% | 99.7% | $6.64 | $125 | $4114 | 47 | $192 | $98 | 22 | 35,454 |
| Angola | AGO | 2,886,471 | 1.87% | 94.8% | $4.41 | $125 | $2336 | 37 | $87 | $61 | 185 | 18,639 |
| Benin | BEN | 1,303,065 | 1.62% | 89.2% | $3.43 | $125 | $1561 | 13 | $21 | $40 | 178 | 8658 |
| Botswana | BWA | 197,961 | 0.53% | 99.2% | $11.56 | $125 | $8029 | 3 | $20 | $202 | 3 | 2175 |
| Burkina Faso | BFA | 2,579,791 | 1.40% | 69.7% | $2.73 | $125 | $1003 | 23 | $23 | $26 | 880 | 15,304 |
| Burundi | BDI | 1,672,818 | 1.80% | 65.4% | $1.80 | $125 | $259 | 13 | $3 | $7 | 550 | 8363 |
| Cameroon | CMR | 2,483,897 | 1.50% | 94.5% | $3.76 | $125 | $1816 | 29 | $53 | $47 | 196 | 18,757 |
| Central African Republic | CAF | 579,823 | 1.90% | 73.0% | $2.18 | $125 | $559 | 5 | $3 | $14 | 227 | 4420 |
| Chad | TCD | 2,156,975 | 2.05% | 68.9% | $2.43 | $125 | $761 | 18 | $14 | $20 | 64 | 1086 |
| Democratic Republic of the Congo | COD | 10,033,279 | 2.10% | 55.9% | $2.28 | $125 | $638 | 101 | $65 | $61 | 5579 | 66,565 |
| Congo | COG | 471,271 | 1.40% | 97.1% | $4.51 | $125 | $2419 | 6 | $15 | $72 | 20 | 3719 |
| Cote d'Ivoire | CIV | 2,653,989 | 1.51% | 85.2% | $5.02 | $125 | $2818 | 29 | $82 | $17 | 569 | 20,272 |
| Djibouti | DJI | 76,593 | 0.66% | 89.6% | $6.09 | $125 | $3676 | 1 | $4 | $92 | 25 | 1274 |
| Egypt | EGY | 11,162,321 | 0.84% | 99.3% | $6.79 | $125 | $4236 | 110 | $465 | $105 | 113 | 88,622 |
| Equatorial Guinea | GNQ | 124,311 | 1.29% | 97.3% | $13.10 | $125 | $9247 | 2 | $15 | $239 | 4 | 681 |
| Eritrea | ERI | 377,047 | 1.05% | 74.5% | $2.34 | $125 | $689 | 4 | $3 | $18 | 366 | 7571 |
| Eswatini | SWZ | 139,161 | 0.87% | 96.7% | $7.26 | $125 | $4606 | 1 | $6 | $120 | 8 | 1319 |
| Ethiopia | ETH | 15,054,111 | 1.41% | 77.9% | $2.77 | $125 | $1031 | 127 | $131 | $27 | 2771 | 66,041 |
| Gabon | GAB | 138,851 | 1.28% | 99.5% | $12.49 | $125 | $8760 | 2 | $21 | $222 | 1 | 1465 |
| Gambia | GMB | 227,585 | 1.26% | 88.3% | $2.62 | $125 | $913 | 3 | $2 | $24 | 37 | 1645 |
| Ghana | GHA | 2,945,235 | 0.91% | 97.4% | $4.83 | $125 | $2672 | 34 | $90 | $68 | 112 | 22,867 |
| Guinea | GIN | 1,492,074 | 1.31% | 73.3% | $3.09 | $125 | $1283 | 15 | $19 | $34 | 515 | 10,130 |
| Guinea Bissau | GNB | 211,493 | 1.25% | 75.5% | $2.59 | $125 | $888 | 2 | $2 | $23 | 72 | 1544 |
| Kenya | KEN | 6,210,954 | 1.01% | 96.1% | $4.23 | $125 | $2193 | 59 | $129 | $57 | 523 | 70,391 |
| Lesotho | LSO | 258,228 | 0.26% | 91.8% | $3.08 | $125 | $1275 | 2 | $3 | $31 | 36 | 2292 |
| Liberia | LBR | 494,532 | 1.19% | 69.2% | $2.40 | $125 | $736 | 6 | $4 | $19 | 226 | 3854 |
| Libya | LBY | 446,788 | 0.35% | 99.9% | $9.74 | $125 | $6577 | 7 | $47 | $160 | 2 | 5667 |
| Malawi | MWI | 2,572,937 | 1.67% | 85.2% | $2.36 | $125 | $702 | 21 | $15 | $18 | 484 | 17,250 |
| Mali | MLI | 2,317,774 | 1.91% | 77.0% | $2.73 | $125 | $1003 | 23 | $23 | $26 | 248 | 5665 |
| Mauritania | MRT | 431,807 | 1.52% | 92.1% | $3.84 | $125 | $1883 | 5 | $10 | $48 | 47 | 3162 |
| Morocco | MAR | 3,028,220 | 0.04% | 98.5% | $6.27 | $125 | $3821 | 39 | $148 | $92 | 80 | 27,846 |
| Mozambique | MOZ | 3,573,783 | 1.65% | 67.2% | $2.16 | $125 | $547 | 35 | $19 | $14 | 1326 | 21,244 |
| Namibia | NAM | 236,440 | 0.61% | 99.1% | $7.96 | $125 | $5168 | 3 | $14 | $133 | 4 | 2433 |
| Niger | NER | 3,403,079 | 2.99% | 58.8% | $2.29 | $125 | $650 | 28 | $18 | $17 | 1169 | 14,935 |
| Nigeria | NGA | 20,493,108 | 1.21% | 92.7% | $4.34 | $125 | $2278 | 228 | $519 | $57 | 1637 | 117,534 |
| Rwanda | RWA | 1,726,921 | 1.44% | 89.5% | $2.62 | $125 | $911 | 14 | $13 | $23 | 115 | 5725 |
| Senegal | SEN | 1,691,043 | 1.63% | 92.7% | $3.68 | $125 | $1755 | 19 | $33 | $45 | 168 | 12,086 |
| Sierra Leone | SLE | 874,255 | 0.99% | 71.1% | $2.18 | $125 | $564 | 9 | $5 | $14 | 301 | 5484 |
| Somalia | SOM | 1,768,684 | 1.91% | 55.2% | $2.09 | $125 | $487 | 18 | $9 | $13 | 1154 | 13,569 |
| South Africa | ZAF | 4,630,640 | 0.15% | 99.1% | $11.09 | $125 | $7643 | 62 | $475 | $190 | 42 | 23,718 |
| South Sudan | SSD | 1,323,559 | 1.20% | 68.9% | $1.97 | $125 | $398 | 12 | $5 | $10 | 41 | 1603 |
| Sudan | SDN | 5,139,412 | 1.49% | 86.5% | $2.52 | $125 | $835 | 48 | $40 | $21 | 163 | 6347 |
| United Republic of Tanzania | TZA | 7,229,829 | 1.73% | 93.3% | $3.03 | $125 | $1241 | 67 | $83 | $28 | 345 | 27,022 |
| Togo | TGO | 902,804 | 1.35% | 90.2% | $2.84 | $125 | $1084 | 9 | $10 | $101 | 102 | 5488 |
| Tunisia | TUN | 929,026 | 0.01% | 99.7% | $6.86 | $125 | $4288 | 12 | $53 | $25 | 5 | 7282 |
| Uganda | UGA | 5,650,640 | 1.56% | 91.4% | $2.65 | $125 | $938 | 51 | $48 | $32 | 133 | 8132 |
| Zambia | ZMB | 2,026,333 | 1.55% | 89.6% | $3.01 | $125 | $1225 | 21 | $25 | $32 | 48 | 2425 |
| Zimbabwe | ZWE | 1,817,317 | 1.23% | 80.7% | $3.86 | $125 | $1898 | 16 | $30 | $49 | 410 | 11,195 |

Precise numbers of this table are in Supplementary Data 9. Supplementary Data 1 is the input file for the simulations, which includes the bounds of different inputs. Supplementary Data 6 contains the calculation of the GDP contribution. Supplementary Data 7 contains the calculation for the dog population using Knobel et al.[27] data and Supplementary Data 8 contains Hampson's data on the PEP reach, number of deaths and exposed[2].

PEP ("all PEP" profile), all countries collaborate to realize a mass dog vaccination program ("all VAC" profile), and finally, the case where only the studied country mass vaccinates dogs meaning that we have a pathogen reintroduction in the midterm. For the last combination, we have 48 "one VAC" profiles (one per country), which we summarize as one column by keeping only the relative gains or losses, as only the studied country will have non-null results in this profile (see Table 1).

By combining all elements, we have a one-to-one correspondence between different inter- and intra-countries choices and associated payoffs. This mapping allows us to compare the strategy profiles and highlight some special ones, like the one where all countries have the best expected payoffs under the assumption that every country is self-interested and cautious (Nash equilibrium), or the one corresponding to the profile with the highest possible gains in total for all countries (Pareto optimal solution), which is in our case the cooperative solution. The mathematical (game) definition and proofs are in Supplementary Information 2.

### Inclusion and ethics statement

All authors have made essential contributions to the conceptualization, data analysis and writing of this study. As a desk study using publicly available data without including data collection on humans or animals, we consider that there are no ethical issues with this work.

### Reporting summary

Further information on research design is available in the Nature Portfolio Reporting Summary linked to this article.

## Data availability

The data and a description of its sources are available as Supplementary Information/Data to this manuscript. There are no accession codes or unique identifiers. Weblinks for publicly available datasets are provided. There is no restriction on data availability. Supplementary Data 1 contains for every country the data used for the simulations; Supplementary Data 2 contains the strategy analysis for every country; Supplementary Data 3 contains the Sobol indices; Supplementary Data 4 contains the neighboring countries matrix; Supplementary Data 5 contains the minimal distance matrix; Supplementary Data 6 contains the GDP data for every country; Supplementary Data 7 contains the dog population data for every country; Supplementary Data 8 contains the human rabies disease burden data for every country. Supplementary Data 9 contains the precise numbers of Table 2 and some additional information.

## Code availability

We have published the code and annotated it. The code is available on the public GitHub repository (https://github.com/SwissTPH/Game_theory_rabies) and the associated DOI on Zenodo (10.5281/zenodo.8208837). We also made public the results generated during our simulations, this can also be used to replicate the figures and the results (Link to the files: https://drive.switch.ch/index.php/s/Xcwax2wn6uj4bLY).

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

## Acknowledgements

We thank Maria Zinsstag to point to Nowak and Coakley[18], which inspired a game theory of One Health. Calculations were performed at sciCORE (http://scicore.unibas.ch/) scientific computing center at University of Basel. A Dimov is funded by the Swiss National Science Foundation Project CRSII5_202257. Publication cost are supported by Swiss National Science Foundation as part of the project 310030_204360. Otherwise there was no funding for this work.

## Author contributions

J.Z. and A.B. designed the study. A.B. collected the data and did the modeling and data analysis, he wrote the first draft of the original draft paper. A.D. recalculated the model and the sensitivity using Sobol indices and complemented the manuscript and figures. N.C. supervised the mathematical modeling, interpreted the data and contributed to writing. G.F. supervised the game theoretical economic analysis, interpreted the data and contributed to the writing. B.B. provided the African collaboration and coordination expertise, J.Z. is the corresponding author, did review editing, data interpretation and provided rabies epidemiological expertise.

## Competing interests

The authors declare no competing interests.
