## [Peer Review File · Nature Communications]

Benefit-cost analysis of coordinated strategies for control of rabies in AfricaREVIEWER COMMENTS

Reviewer #1 (Remarks to the Author):

General comments:

A. Remove the economics and modeling related jargon: Remember your audience. I assume that your prime intended audience is practicing public health officials and policy makers. Many are unlikely to have taken advanced economics or modeling courses. Examples of the type of jargon that I strongly recommend that you remove include:

- i. "deterministic meta-population model"
- ii. "game theoretic setting" (I note that Appendix 2 contains a very technical description of game theory – but I think that this will not help most of your intended audience).
- iii. Nash equilibrium (only economists use this/ are familiar with this)
- iv. Pareto-optimal

B. Could this be independently replicated by somebody with sufficient skills and interest? I have my doubts that this study it could indeed be independently replicated. One reason for my concerns regarding ability to replicate is that I found it hard to understand the input variables (see specific comment #3).

Specific comments:

1> Introduction: Lines 91-98: Rewrite to remove jargon: Your intended audience is unlikely to be familiar with terms such as "game theoretical framework." The explanation given (Line 93-94) ". . . engage in strategic interactions that determine subsequent health, financial and environmental outcomes" is unlikely to help such an audience understand game theory and its potential to answer the question posed in the paper. And do not assume that the intended audience will find the time-and-motivation to seek out and read reference (#19).

Strongest recommendation: Rewrite this last paragraph of the Introduction to emphasize to the reader what they would gain from reading your paper (I provide a brief example below).

Example of re-written last paragraph of Introduction section: "In this paper we present the economic impact of African countries coordinating various combinations of human post-exposure prophylaxis and dog rabies vaccination programs. The goal of such inter-country coordination is to reduce human deaths due to dog rabies beyond what could be achieved if countries acted solely on their own."

2> Remove claims of being first: Lines 94-96 ". . . has, to our knowledge, never been used in OH to investigate the impact of interventions . . ."

You may or may not be the first to use game theory to analyze a One Health problem. BUT, does it matter if you are the first? Surely your contribution "speaks for itself" regardless of how many publications already exist on this topic? Claims of "being first" are distractions.

Recommendation: Remove all references to "being the first" and equivalent statements.

3> Methods: Lacking Table in main text showing inputs, values and sources: I appreciate that the authors have, in Appendix 0, Table 3 ("Notations used for the SEIR model for rabies in the dog population.") provided some input values. Nonetheless, I believe that the intended audience will often struggle to understand what went into the model (different to type of model).

Note that providing the Beta function(as in the Appendix) is unlikely to help many readers readily understand something such as assumed number of dogs in a given country.

Missing costs? I could not readily find a Table listing the inputted/ assumed costs lines 109-11). The intended audience will likely be very interested in quickly understanding what you assumed as the costs of PEP, dog rabies vaccination etc.

Similarly, readers will want to know what exactly what “. . . GDP per capita was then utilized as a proxy for the economic contribution of an individual in a given year.” (Lines 125-126). Do assume that readers will rush to consult references #7 and #8.

By the way – I note that Reference #8 is not the actual World Bank database on GDP per capita mentioned in Line 127.

Journal limits on the number of figures/ tables: I recognize that Nature Communications limits to 4 the number of Tables/ Figures in the main text. But, readers need readily accessible information (i.e., in the main text) about input values to help them absorb and “orientate” themselves so that they can understand and appreciate the Results. Thus, the authors will need to move an existing Figure or Table in the main text to the Appendix and produce a new Table for the main text listing input values. Some long input value Tables, such as GDP per capita and number of dogs, can be placed in the Appendix.

ESSENTIAL RECOMMENDATION: I suggest that the authors move Figure 2 to the Appendix and put in a new Table listing input values, as described above.

4> Methods: Game theory: Lines 132 -137: I doubt, based on the description of game theory in Lines 132-137, that the intended audience will really understand.

What is missing: A concise sentence describing what you attempted to do with game theory. Another sentence describing in basic, non-jargon laden terms, what is “game theory” – WITHOUT the use of terms such as Nash equilibrium and pareto optimal (terms that typically only economists use).

For example; “To define what may be the best mix of increased use of PEP and dog rabies vaccination, along with cross-country collaborations for dog rabies vaccination programs, we considered several strategies. The strategies examined include . . . [Move Lines 199 – 120 to here – it is a description of Methods, sitting in the results section c.f., Point #5, below]”

5> Methods in the Results section: Lines 199-219 (Excluding Table 1 – which presents Results). This Section, labeled “Strategy analysis,” is really Methods. It should be move – as well as re-writing the material so it will be clearer to non-economists (c.f., point #4 above).

Reviewer #2 (Remarks to the Author):

The authors performed a cost-benefit analysis to discuss dog mass vaccination campaigns focusing at the elimination of dog-mediated rabies. They make a strong point for integrated actions between different areas (such as veterinary and human health), and between countries. The analysis and presentation of data are sound, and conclusions are based on presented results. The adoption of the One Health Approach will definitely improve control measures, but (as discussed by the authors) will face several problems during planning and implementation.

I have only two points which may be considered in a revised version of the manuscript:

1. Line 109 - the authors claimed to have used "all currently available data". It would be nice to read some more details on data sources, available variables, quality and completeness of data etc.
2. The results section present for the 48 countries breakeven points for vaccination programs, the absolute and relative gains etc. However, it would be nice for the reader to have also a presentation of the current rabies burden (no. of cases) by country - this may included in the figures or tables.

Letter of revision: Answers to reviewer queries are in red.

REVIEWER COMMENTS

Reviewer #1 (Remarks to the Author):

General comments:

A. Remove the economics and modeling related jargon: Remember your audience. I assume that your prime intended audience is practicing public health officials and policy makers. Many are unlikely to have taken advanced economics or modeling courses. Examples of the type of jargon that I strongly recommend that you remove include:

- i. “deterministic meta-population model”
- ii. “game theoretic setting” (I note that Appendix 2 contains a very technical description of game theory – but I think that this will not help most of your intended audience).
- iii. Nash equilibrium (only economists use this/ are familiar with this)
- iv. Pareto-optimal

Answer: Thank you for this comment. The prime intended audience of this paper are surely public health officials and policy makers. But, we trust that this article will also be of interest to epidemiologists, disease modelers and health economists. Therefore, we removed the jargon terms in the main text and leave them in brackets to assure that non-specialist readers understand and academic readers appreciate what we write.

B. Could this be independently replicated by somebody with sufficient skills and interest? I have my doubts that this study it could indeed be independently replicated. One reason for my concerns regarding ability to replicate is that I found it hard to understand the input variables (see specific comment #3).

Answer: We have published the code and annotated it (Link to the public GitHub repository: https://github.com/SwissTPH/Game_theory_rabies). We also made public the results generated during our simulations, this can also be used to replicate the figures and the results (Link to the files: <https://drive.switch.ch/index.php/s/Xcwax2wn6uj4bLY>). Hence, we assume that the reader with sufficient skills will be able to replicate our study.

Specific comments:

1> Introduction: Lines 91-98: Rewrite to remove jargon: Your intended audience is unlikely to be familiar with terms such as “game theoretical framework.” The explanation given (Line 93-94) “. . . engage in strategic interactions that determine subsequent health, financial and environmental outcomes” is unlikely to help such an audience understand game theory and its potential to answer the question posed in the paper. And do not assume that the intended audience will find the time-and-motivation to seek out and read reference (#19).

Strongest recommendation: Rewrite this last paragraph of the Introduction to emphasize to the reader what they would gain from reading your paper (I provide a brief example below).

Example of re-written last paragraph of Introduction section: “In this paper we present the economic impact of African countries coordinating various combinations of human post-exposure prophylaxis and dog rabies vaccination programs. The goal of such inter-country coordination is to reduce human deaths due to dog rabies beyond what could be achieved if countries acted solely on their own.”

Answer: We have revised the last paragraph of the introduction.

2> Remove claims of being first: Lines 94-96 “. . . has, to our knowledge, never been used in OH to investigate the impact of interventions . . .”

You may or may not be the first to use game theory to analyze a One Health problem. BUT, does it matter if you are the first? Surely your contribution “speaks for itself” regardless of how many publications already exist on this topic? Claims of “being first” are distractions.

Recommendation: Remove all references to “being the first” and equivalent statements.

Answer: We removed all mentions of being first.

3> Methods: Lacking Table in main text showing inputs, values and sources: I appreciate that the authors have, in Appendix 0, Table 3 (“Notations used for the SEIR model for rabies in the dog population.”) provided some input values. Nonetheless, I believe that the intended audience will often struggle to understand what went into the model (different to type of model).

Note that providing the Beta function (as in the Appendix) is unlikely to help many readers readily understand something such as assumed number of dogs in a given country.

Missing costs? I could not readily find a Table listing the inputted/ assumed costs (lines 109-11). The intended audience will likely be very interested in quickly understanding what you assumed as the costs of PEP, dog rabies vaccination etc.

Answer: We added a table with the input data of the model, and we added mentions to all relevant appendices with original data or calculations.

Similarly, readers will want to know what exactly what “. . . GDP per capita was then utilized as a proxy for the economic contribution of an individual in a given year.” (Lines 125-126). Do assume that readers will rush to consult references #7 and #8.

Answer: We added this information to the table with the input data.

By the way – I note that Reference #8 is not the actual World Bank database on GDP per capita mentioned in Line 127.

Answer: Thank you for pointing it out, there was a problem during the formatting. The references were updated and corrected.

Journal limits on the number of figures/ tables: I recognize that Nature Communications limits to 4 the number of Tables/ Figures in the main text. But, readers need readily accessible information (i.e., in the main text) about input values to help them absorb and “orientate” themselves so that they can understand and appreciate the Results. Thus, the authors will need to move an existing Figure or Table in the main text to the Appendix and produce a new Table for the main text listing input values. Some long input value Tables, such as GDP per capita and number of dogs, can be placed in the Appendix.

ESSENTIAL RECOMMENDATION: I suggest that the authors move Figure 2 to the Appendix and put in a new Table listing input values, as described above.

Answer: We did as you suggested, we moved the figure 2 to the appendix and added the table with the input data.

4> Methods: Game theory: Lines 132 -137: I doubt, based on the description of game theory in Lines 132-137, that the intended audience will really understand.

What is missing: A concise sentence describing what you attempted to do with game theory. Another sentence describing in basic, non-jargon laden terms, what is “game theory” – WITHOUT the use of terms such as Nash equilibrium and pareto optimal (terms that typically only economists use).

For example; “To define what may be the best mix of increased use of PEP and dog rabies vaccination, along with cross-country collaborations for dog rabies vaccination programs, we considered several strategies. The strategies examined include . . . [Move Lines 199 – 120 to here – it is a description of Methods, sitting in the results section c.f., Point #5, below]”

Answer: The paragraph was reworked. As we mentioned in a previous answer, we use descriptive terms and keep some of the technical terms for the economic or modelling specialists in brackets.

5> Methods in the Results section: Lines 199-219 (Excluding Table 1 – which presents Results). This Section, labeled “Strategy analysis,” is really Methods. It should be moved – as well as re-writing the material so it will be clearer to non-economists (c.f., point #4 above).

Answer: The section “Lines 199-219” was moved to the Methods and reworked. Some parts were added to explain more the meaning of the economical jargon.

Reviewer #2 (Remarks to the Author):

The authors performed a cost-benefit analysis to discuss dog mass vaccination campaigns focusing at the elimination of dog-mediated rabies. They make a strong point for integrated actions between different areas (such as veterinary and human health), and between countries. The analysis and presentation of data are sound, and conclusions are based on presented results. The adoption of the One Health Approach will definitely improve control measures, but (as discussed by the authors) will face several problems during planning and implementation.

I have only two points which may be considered in a revised version of the manuscript:

1. Line 109 - the authors claimed to have used "all currently available data". It would be nice to read some more details on data sources, available variables, quality and completeness of data etc.

Answer: We added a table with the input data of the model, and we added mentions to all relevant appendices with original data or calculations.

2. The results section present for the 48 countries breakeven points for vaccination programs, the absolute and relative gains etc. However, it would be nice for the reader to have also a presentation of the current rabies burden (no. of cases) by country - this may included in the figures or tables.

Answer: We added some complementary information on the burden in the table with the input data and to the Appendix 4bis.

REVIEWERS' COMMENTS

Reviewer #1 (Remarks to the Author):

My comments have been adequately addressed - thank you. No further comments.